# Assessment of knowledge, attitude, and practice of disposing and storing unused and expired medicines among the communities of Kathmandu, Nepal

Nisha Jha[1]*, Sajala Kafle[1], Shital Bhandary[2], Pathiyil Ravi Shankar[3]

1 Department of Clinical Pharmacology and Therapeutics, KIST Medical College, Lalitpur, Nepal,
2 Department of Community Health Sciences and School of Public Health, Patan Academy of Health Sciences, Patan, Lalitpur, Nepal, 3 IMU Centre for Education, International Medical University, Kuala Lumpur, Malaysia

* nishajha32@gmail.com

## Abstract

### Background

Unused medicines can be stored by many people at their places of residence and houses for later use. This study evaluated knowledge, and attitudes regarding unused and expired medicines and explored medicine storage and disposal practices among selected households in the Kathmandu valley, Nepal.

### Method

A cross-sectional study with a two-stage cluster survey design was done using a semi-structured questionnaire from April to October 2021. The sample size (total number of households) after adjusting for design effect and non-response rate was 210 and the study population was the household heads. Simple random sampling was done to select clusters during the first stage and systematic random sampling to select households during the second stage. Descriptive statistics and t-test/one-way ANOVA were used to compare the respondents' average knowledge scores. Practice variables were presented using frequency distribution.

### Results

Around half the respondents were from the Kathmandu district, nearly 20% were from Bhaktapur and 30% were from Lalitpur. Nearly two-thirds were male and about 25% had a bachelor's degree. Nearly 90% of respondents agreed that storage of excess medicines at home may promote self-medication. Similarly, 97.6% of respondents agreed there is a lack of adequate information on the safe disposal of unused medicines. The majority [125 (59.5%)] of participants always checked the expiry date of medicines. The safe methods of medicine disposal were not known by 137 (65.2%) participants. Throwing in a dustbin was the preferred method of expired medicine disposal.

**Data Availability Statement:** The data related to the study is available from: https://figshare.com/articles/dataset/Storage_and_medicine_disposal_

by_households_in_Kathmandu_Valley_Nepal_/
17128469.

**Funding:** The study received funding of Nepalese
rupees 125000 from the Nepal Health Research
Council. The funders had no role in study design,
data collection, and analysis, decision to publish, or
preparation of the manuscript.

**Competing interests:** This does not alter our
adherence to PLOS ONE policies on sharing data
and materials.

## Conclusion

The level of knowledge and practice of disposing of unused and expired medicines requires
improvement. Educational interventions may help improve awareness further. Creating a
chart summarizing disposal procedures of common medicines is important. Similar studies
in other regions are required.

## Introduction

Many medicines have an expiry date and should not be used by patients after that date.
Unused and leftover medicines can be stored by many people in their places of residence and
houses for later use. Similarly, the medicines remaining after the patient's recovery or after the
occurrence of any adverse drug reactions can also be stored. All of these can contribute to the
wastage of medicines [1,2].

Medicines are very important for enhancing the quality of life and managing disease conditions [3]. Proper and safe disposal of medicines is important as inappropriate and unsafe disposal can have harmful effects with traces of disposed medicines persisting in the environment
and the water systems [4]. Safety precautions should be taken with the leftover medicines to
avoid accidental and hazardous effects, especially among children, the elderly, and pregnant
women. Misuse and abuse can also occur because of the leftover medicines in the households
[5].

Safe disposal of medicines is an important issue. There is a lack of knowledge about safe
and proper medicine disposal practices among consumers [6,7]. The common method used to
dispose of medicines are throwing in the garbage, flushing down the toilets, and sharing with
relatives and friends [8–10]. Programs for safe medicine disposal exist in developed countries
but are lacking in developing countries like Nepal. Pharmacists are also unaware of the safe
medicine disposal in developing countries [10]. The systems for medicine take-back are well
established in developed countries like Canada, Australia, the United States, and Sweden
[11,12]. Guidelines have been framed by the World Health Organization (WHO) for waste
management in the pharmaceutical sector and burning the waste in the open-air is not recommended due to its possible toxicity to the environment [13].

In Nepal, there is evidence of poor awareness among the pharmacists working in community pharmacies about the safe disposal of medicines [14]. The provision of color coding and
segregation of waste is missing in the pharmacies. Improper medicine disposal practices were
seen in Pokhara, Nepal [13]. Proper storage of medicines is very important to maintain their
therapeutic efficacy and prevent degradation. Improper disposal of medicines can lead to the
presence of different types of medicines like hormones, tranquilizers, antiepileptics, antidepressants, and drugs against cancer in the water systems. Different types of toxic substances
can be released along with gases that can affect human health [3,15–18].

Studies exploring the knowledge about the safe storage and disposal of medicine have not
been carried out in Nepal. Hence, this study was conducted in the Kathmandu valley. The valley is divided into the three districts of Kathmandu, Lalitpur, and Bhaktapur. The findings of
this study can be important to policymakers, health professionals, pharmaceutical companies,
and the community in general.

The study evaluated knowledge, and attitude regarding unused and expired medicines and
explored medicine storage and disposal practices among selected households in the Kathmandu valley, Nepal.

## Methodology

### Study area and period

The study was conducted in the Kathmandu Valley from April to October 2021. Kathmandu is the capital and the most populous city of Nepal. Healthcare is most developed in Kathmandu, and the valley is home to some of the best hospitals and clinics in the country. The area of Kathmandu valley is 665 square kilometers with a population of 1.442 million. The number of health facilities in the district is 2320 which is a high proportion of the total number in Nepal (6934) [19,20].

### Study design

This was a cross-sectional study with a two-stage cluster survey design.

The quantitative study was conducted using a semi-structured questionnaire. The variables of age, gender, literacy, occupation, work experience, educational status, monthly income, number of household members, religion, and district of residence were recorded.

### Sampling

There are 21 municipalities in the Kathmandu valley (18 urban municipalities and 3 rural municipalities). One ward (cluster) of each of these municipalities was randomly selected. This was followed by selecting 10 households randomly from each of the 21 selected clusters to obtain a total sample size of 210, similar to the one used during the WHO Expanded Program on Immunization (EPI) survey [21].

Simple Random sampling was used to select the clusters during the first stage and systematic random sampling to select households during the second stage.

There were 40 households selected from Bhaktapur district as it has 4 municipalities only. Kathmandu has 11 and Lalitpur has 6 municipalities so there were 110 samples from Kathmandu and 60 samples from the Lalitpur districts of the Kathmandu valley.

The study population was the household heads (HHH). If the HHH was not available at the time of the visit and two subsequent attempts, then the interview was conducted with the person who was acting as HHH in that household. The research assistant hired and trained for this study identified the ward office and handed over the ethical approval letter from the Nepal Health Research Council. After obtaining administrative approval, he selected 10 households using systematic random sampling and the sampling frame provided by the ward office. He then collected data from these 10 households.

### Sample size calculation

There was one published study from Pokhara (Western Nepal) showing that 50% of the respondents were practicing proper drug disposal [10]. We took this as the reference to calculate the sample size for this study. As a similar study was not done earlier in the Kathmandu valley, we decided to use a margin of error of 10% giving a proper practice range of 40–60% with a 95% confidence interval for the sample size. We used www.openepi.com to calculate the sample size of 97 using these parameters. This sample size was multiplied by a design effect of 2 to adjust the variance introduced due to clusters (random wards) giving a sample size of 193, which was further adjusted with a 5% non-response rate giving the final minimum sample size of 203. As we planned to select an equal number of households from each randomly selected ward from each of the 21 municipalities of the Kathmandu Valley, we decided to randomly select 10 households from each municipality giving the final sample size of 210 for this study. The ward was selected using the "randbetween" function using the number of wards in each

municipality as the frame whereas households were selected randomly from the sampling frame of the selected clusters using a systematic random sampling method. This was a two-stage cluster sampling with random sampling at both stages.

### Data collection tool

A semi-structured questionnaire was designed by the researchers based on the literature and referring to the medicine disposal guidelines from the WHO and other international and national agencies [10,17,18]. The questionnaire had three sections. Section I obtained demographic details regarding age, gender, literacy, educational status, occupation, monthly income, work experience, family size, and district of residence. Section II contained questions regarding knowledge and attitude toward medicine storage and disposal. The knowledge questions were structured as multiple-choice questions (MCQs) whereas attitude questions measured agreement with the statement using a 4-point Likert scale. For the knowledge MCQs, right answers were scored as '1' and wrong answers as '0'. A similar type of scoring has been used in other studies [10,17]. Section III contained questions about the disposal practice of unused and expired pharmaceuticals among the households. Some questions were open-ended to allow respondents to explain the barriers and facilitators of good practices.

The questionnaire was translated into the Nepali language for obtaining valid responses from the participants who cannot understand English. A bi-lingual expert performed a forward translation, and another bi-lingual expert independently performed the backward translation. The original and backward translated questionnaires were discussed by the study team multiple times and corresponding changes were done to the Nepali language questionnaire. The questionnaire was completed during a face-to-face interview by a trained data collector. The Nepali version of the questionnaire was pretested by administering it to 10 households among a non-selected cluster of the Kathmandu district by the research team members and queries raised by the respondents were noted. Necessary changes were made to the questionnaire to establish the face validity.

The finalized Nepali questionnaire was sent to five content experts for content validation. As per their reports, some of the questions were modified and corrected by the study team. A pilot test was done among 20 households, and the Cronbach's alpha obtained was 0.863 for Knowledge and 0.866 for Attitude. Since the practice was measured using a mixed type of questions, internal consistency was not computed for this section.

Knowledge score was calculated by adding the correct (coded as 1) and incorrect (coded as 0) responses to 10 knowledge questions (Q1-Q10) with a possible score between 0 and 10 (maximum score of 10).

This scale was summarized using a histogram and descriptive statistics.

The histogram was found to be "tentatively" bell-shaped, so parametric tests (t-test and one-way ANOVA) were used to compare the knowledge score among different subgroups. T-test was used to compare knowledge scores among subgroups with two categories and one-way ANOVA and used to compare knowledge among subgroups with more than two categories.

The overall mean knowledge score was calculated using simple random.

PSS file by one of the team members. Once all the data were entered, it was checked with the original and any inconsistencies found were corrected as per the unique ID and the corresponding questionnaire by the study team. The final data was saved securely in the principal investigator's personal laptop. A working copy with the de-identified file was used for the data analysis.

Descriptive statistics were used to present the data and t-test/one-way ANOVA tests to compare the average knowledge scores among respondents. Practice variables were presented using frequency distribution among the study population. Analysis of knowledge and attitude scores by municipalities and analysis of practice questions by districts are provided in the appendix. A p-value of less than 0.05 was considered statistically significant.

### Ethical considerations

Ethical approval was obtained from the Ethical Review Board of Nepal Health Research Council, dated 24th March 2021 with a reference number 124/2021P.

## Results

Table 1 shows the background characteristics of the respondents. Around half, 110 (52.4%) of the respondents were from Kathmandu district whereas 40 (19%) respondents were from Bhaktapur, and 60 (28.6%) respondents were from Lalitpur districts.

### Demographic characteristics of the participants

**Knowledge response of the participants.** Table 2 shows the responses to the ten knowledge questions. The complete questionnaire is shown in the Appendix. Nearly all the respondents (97.1%) had correct knowledge of the responsibility of the healthcare professional to improve knowledge of proper disposal of "unused medicine" at the household level. Around 97% of the respondents identified correct responses for inappropriate disposal of medicines in an unauthorized manner causing various harmful effects.

The knowledge scores were significantly different only among respondents with different educational levels (p-value < 0.05). So, a pairwise comparison was done for this variable using Tukey's HSD post hoc test, and a statistically significant difference was observed between respondents with Secondary and University levels of education only (Table 3).

**Analysis of attitude of the participants.** Table 4 shows that nearly 70% of the respondents "strongly agreed" and the remaining 29.5% "agreed" that outreach and awareness programs on the disposal of unused and expired medicines should be initiated.

Similarly, 70% of the respondents "strongly agreed" and the remaining 27.6% "agreed" that there is a lack of adequate information on the safe disposal of unused medicines. Likewise, nearly 65% of the respondents "strongly agreed" and another 34% "agreed" that doctors and health professionals should advise on the safe disposal of unused and expired household medicines.

An attitude score was computed by adding the responses to questions 11 to 21 and it was summarized using a histogram and descriptive statistics. The mean attitude score was 16.3 with a standard deviation of 4.497. This means 95% of the responses were between 16.30 ± 24.50 i.e., 7.30 and 25.30. As the minimum and maximum possible values of the attitude scale were 11 and 44 for these 11 questions measuring attitude, the scale mean was towards the minimum value suggesting a positive attitude towards the medicine storage and disposal mechanisms covered by these 11 questions. The histogram of the attitude scale was not bell-shaped, so the median test was used to compare attitude scores (dependent variable) with other (independent) variables (Table 5).

The median attitude score was significantly different according to the district of residence. So, a pairwise comparison was done, and it revealed a statistically different attitude score between Kathmandu (14) and Lalitpur (17) districts only. Kathmandu and Bhaktapur had a low median attitude scale (positive attitude) than the Lalitpur district.

**Table 1. Demographic characteristics of the study participants.**

| Variables | Number (Percentage) |
|---|---|
| **District** | |
| Kathmandu | 110 (52.4) |
| Bhaktapur | 40 (19.0) |
| Lalitpur | 60 (28.6) |
| **Age (Median = 30 years)** | |
| <30 years | 100 (47.6) |
| >= 30 years | 110 (52.4) |
| **Gender** | |
| Male | 131 (62.4) |
| Female | 79 (37.6) |
| **Literacy status** | |
| Literate | 195 (92.9) |
| Illiterate | 15 (7.1) |
| **Education level** | |
| No formal education and primary level | 24 (11.4) |
| Secondary level | 85 (40.5) |
| University level | 101 (48.1) |
| **Occupation** | |
| No work | 22 (10.5) |
| Daily wage | 108 (51.4) |
| Service | 26 (12.4) |
| Retired | 29 (13.8) |
| Housewife | 16 (7.6) |
| Others | 9 (4.3) |
| **Average monthly income (NPR)** | |
| No income | 32 (15.7) |
| Less than 10000 | 28 (8.8) |
| 10000–20000 | 60 (29.4) |
| 20000–40000 | 72(35.3) |
| 40000–60000 | 22 (10.8) |
| **Number of family members (Median = 5)** | |
| <5 | 85 (40.5) |
| >= 5 | 125 (59.5) |

Note: The working experience variable is not described as 78% of cases had missing information.

**Analysis of the practice of the participants.** As the practice questions were mixed in nature (rating scale, multiple choices, true/false etc.), the practice score was not generated, and the results were presented descriptively using frequency distribution. Results for practice showed that 125 (59.5%) participants always checked the expiry date of medicines. For the statement regarding what do you do with any quantity of purchased medicine remaining unused at your home/hostel, many, 75 (35.7%), participants responded throwing it away in household garbage. Similarly, for taking medications as per a Doctor's/ Pharmacist's advice, about 120 (57.1%) participants responded that they always follow the advice. Practicing self-medication for minor illnesses like fever and headache was done sometimes by 80 (38.1%) respondents. The results regarding the readability of the expiry dates in the medicine dosage forms were responded to as always by 126 (60.0%) respondents.

**Table 2. Correct and incorrect responses for knowledge statements.**

| Question | Correct response N (%) | Incorrect response N (%) |
|---|---|---|
| Q1. 'Expiry date of medicine' means: | 22 (10.5) | 188 (89.5) |
| Q2. The term 'Medicine disposal system' refers to: | 35 (16.7) | 175 (83.3) |
| Q3. The term 'Medicine take back system' for the expired medicines is: | 56 (26.7) | 154 (73.3) |
| Q4. Inappropriate disposal of medicines in an unauthorized manner can cause all except: | 203 (96.7) | 7 (3.3) |
| Q5. The best method for preventing the hazardous effect of unused medicines can be: | 155 (73.8) | 55 (26.2) |
| Q6. The best strategy for preventing the hazardous effect of expired medicines can be: | 116 (55.2) | 94 (44.8) |
| Q7. Healthcare professionals will be responsible for improving the knowledge among households about the proper disposal of 'Unused medicines'? | 204 (97.1) | 6 (2.9) |
| Q8. The responsible person (s) for improving the knowledge among households about the proper disposal of 'Expired medicines' is (are) healthcare professionals. | 54 (25.7) | 156 (74.3) |
| Q9. Most medicines should be stored in: | 58 (27.6) | 152 (72.4) |
| Q10. Storing excess medicines at home may promote self-medication: | 191 (91.9) | 19 (9) |

The safe methods of medicine disposal were not known by 137 (65.2%) participants. Most respondents, 121 (57.6%) always used all prescribed medications as recommended. Eighty eight (41.9%) participants said that they rarely store unused medications. For liquid medicines, 144 (68.6%) participants never emptied the liquid in the toilet and recycled the glass bottle. Similarly, advising family members on proper medicine disposal was rarely carried out by 71 (33.8%) participants. of those responding 156 (74.3%) thought that the hazardous effect of unused and expired medicines can be minimized or controlled by providing proper guidance to the consumers. Excess quantity supplied was quoted as a reason for leftover or unused medicines at home by 69 (32.9%) respondents. Other reasons mentioned were storing the medicines for future use, sharing the medicines with their friends and relatives, and addressing the need for medicines in emergencies.

Throwing in a dustbin was the preferred method of expired medicine disposal by 135 (64.3%) respondents. Other comments were to burn the remaining medicines for their disposal. The best location to educate the community about the appropriate disposal of unused medicine was identified as the pharmacy while dispensing by 89 (42.4%) respondents. Other methods mentioned were the use of social media, the internet, and mass media to educate people about safe disposal practices.

Unused medicines were kept at home for future use by 141 (37.1%) respondents. The fact that safe disposal of medicines is necessary to prevent adverse reactions was mentioned by 96 (45.7%) participants. Other comments were to prevent environmental pollution and protect people from possible toxic effects of medicines.

## Discussion

Medicine disposal is an important issue for the safety of our environment and human health. Medicines eventually reach the consumers and the patients. Not all medicines that reach consumers are consumed and some are leftover and stored in the houses of the consumers. WHO reports that more than half of the medicines are not properly prescribed and dispensed. This can lead to unwanted and unnecessary storage of medicines at houses [22]. In a developing

**Table 3. Comparison of knowledge scores among different subgroups of respondents.**

| Variable | Mean ± SD | p-value |
|---|---|---|
| **District** | | 0.473[a] |
| Kathmandu | 6.57 ± 1.20 | |
| Lalitpur | 6.78 ± 1.21 | |
| Bhaktapur | 6.75 ± 1.03 | |
| **Age** | | 0.875[b] |
| < 30 years | 6.68 ± 1.13 | |
| >= 30 years | 6.65 ± 1.22 | |
| **Gender** | | 0.986[b] |
| Male | 6.66 ± 1.20 | |
| Female | 6.67 ± 1.13 | |
| **Literacy status** | | 0.254 |
| Literate | 6.69 ± 1.16 | |
| Illiterate | 6.33 ± 1.29 | |
| **Education level** | | **0.015[a]** |
| None and Primary | 5.54 ± 1.31 | |
| Secondary | 5.51 ± 1.51 | |
| University | 5.97 ± 1.06 | |
| **Occupation** | | 0.311 |
| No work | 6.68 ± 1.39 | |
| Daily wage | 6.76 ± 1.20 | |
| Service | 6.62 ± 1.10 | |
| Retired | 6.38 ± 1.02 | |
| Housewife | 6.31 ± 1.01 | |
| Other | 7.22 ± 1.09 | |
| **Monthly income** | | 0.429[a] |
| No income | 6.81 ± 1.26 | |
| < 10,000 | 6.17 ± 1.20 | |
| 10,000–20,000 | 6.65 ± 1.21 | |
| 20,000–40,000 | 6.69 ± 1.16 | |
| 40,000–60,000 | 6.73 ± 0.99 | |
| **Number of family members** | | 0.498 |
| < 5 | 6.60 ± 1.20 | |
| >= 5 | 6.71 ± 1.20 | |

Note:

[a] = One-way ANOVA,

[b] = Independent samples student's t-test.

country like Nepal, out-of-pocket expenditure is commonly incurred for buying medicines and managing health conditions. WHO estimates this expenditure as 70% of out of pocket expenditure [23].

In this study, knowledge scores were less among the participants from Kathmandu district, age more than 30 years, males, illiterates, participants having none or primary level of education, housewives, people having a monthly income less than 10000 NPR (80.65 USD) and with less than 5 family members. Similarly, the attitude scores were lower for participants from Kathmandu and Bhaktapur districts, those aged more than 30 years, males, illiterate, with secondary educational level, daily wage earners, service, and housewife groups, having monthly

**Table 4. Distribution and summary statistics for the attitude questions.**

| Statement | Strongly Agree n (%) | Agree n (%) | Disagree n (%) | Strongly Disagree n (%) | Mean Score ± SD (Range: 1–4) |
|---|---|---|---|---|---|
| Q11: Unused medicines should not be thrown into dustbins | 68 (32.4) | 62 (29.5) | 61 (29.0) | 19 (9.0) | 2.35 ± 1.21 |
| Q12: Burning of expired medicines can release toxic substances which can be inhaled by the people | 145 (69.0) | 58 (27.6) | 1 (0.5) | 6 (2.9) | 1.38±0.65 |
| Q13: Unsafe practices of medicines disposal can pollute the environment | 140 (66.7) | 65 (31.0) | 3 (1.4) | 2 (1.1) | 1.37±0.57 |
| Q14: There should be a public awareness program about the harmful effects of improper medicine disposal practices | 143 (68.1) | 60 (28.6) | 2 (1) | 5 (2.4) | 1.38±0.64 |
| Q15: Community pharmacists have an important role in mitigating the problem of improper medicine disposal practices | 105 (50.0) | 91 (43.3) | 2 (1) | 12 (5.7) | 1.65±0.78 |
| Q16: Unused and expired medicines present potential risks at home | 134 (63.8) | 72 (34.3) | 2 (1) | 2 (1) | 1.39±0.57 |
| Q17: There is a lack of adequate information on the safe disposal of unused medicines | 147 (70.0) | 58 (27.6) | 1 (0.5) | 4 (1.9) | 1.35±0.57 |
| Q18: Children are more vulnerable to the risks associated with unused and expired household medicines | 142 (67.6) | 65 (31) | 1 (0.5) | 2 (1) | 1.35±0.55 |
| Q19: Doctors and health professionals should provide advice on the safe disposal of unused and expired household medicines | 136 (64.8) | 71 (33.8) | 3 (1.4) | 0 (0) | 1.39±0.57 |
| Q20: Drug take-back programs for unused and expired medicines should be mandatory | 136 (64.8) | 65 (31) | 2 (1) | 7 (3.3) | 1.43±0.69 |
| Q21: Outreach and awareness programs about how to dispose of unused and expired medicines should be initiated | 146 (69.6) | 62 (29.5) | 1 (0.5) | 1 (0.5) | 1.32±0.51 |

income between NPR 20000–40000 and 40000–60000 (162–323 and 323–484 USD) and participants with more than 5 family members. These groups of people should be prioritized for interventions for safe medicine disposal practices.

## Knowledge

There are few studies about the roles of pharmacists in educating patients about safe medicine disposal but, their knowledge about safe disposal practices may not be good enough to provide consistent information to the patients [24]. The safe methods of medicine disposal were not known by 137 (65.2%) participants. Similarly, the medicine disposal system was also incorrectly answered by 83.3% of people. 'Medicine take back system' for the expired medicines was also incorrectly responded to by 73.3% of people. Another study mentioned that only 13.3% of respondents were aware of the proper medicine disposal methods and only 1% of participants knew about returning the medicines to the pharmacies for the medicine take-back systems [25]. A study revealed that about half (50.1%) of the participants had a good knowledge concerning the disposal of unused and expired medicines [26].

## Attitude

Our results showed that about 70% of the respondents "strongly agreed" and the remaining "agreed" that outreach and awareness programs on the disposal of unused and expired medicines should be initiated. This is very similar to another study, where almost 70.8% of participants knew about the waste related to medicines and 59.2% checked the expiry dates of the medicines. This study also showed that 53,5% of the respondents strongly agreed that unused medicine can be a great risk at home [27].

**Table 5. Comparison of attitude scores among different subgroups of respondents.**

| Variable | Median± IQR | p-value |
|---|---|---|
| **District** | | **0.026**[c] |
| Kathmandu | 14.00 ± 5.25 | |
| Lalitpur | 17.00 ± 8.00 | |
| Bhaktapur | 14.00 ± 6.75 | |
| **Age** | | 0.797 |
| < 30 years | 14.50 ± 7.00 | |
| >= 30 years | 14.00 ± 8.00 | |
| **Gender** | | 0.544 |
| Male | 14.00 ± 6.00 | |
| Female | 15.00 ± 9.00 | |
| **Literacy status** | | 0.515 |
| Literate | 14.00 ± 7.00 | |
| Illiterate | 17.00 ± 11.00 | |
| Education level | | 0.194 |
| None and Primary | 14.5±10 | |
| Secondary | 14 ±7 | |
| University | 15 ±8 | |
| **Occupation** | | 0.194 |
| No work | 15.00 ± 9.00 | |
| Daily wage | 14.00 ± 9.00 | |
| Service | 14.00 ± 6.00 | |
| Retired | 16.00 ± 7.00 | |
| Housewife | 14.00 ± 2.00 | |
| Other | 16.00 ± 8.00 | |
| **Monthly income** | | 0.087 |
| No income | 15.00 ± 8.00 | |
| < 10,000 | 16.50 ± 9.00 | |
| 10,000–20,000 | 15.00 ± 9.00 | |
| 20,000–40,000 | 14.00 ± 6.00 | |
| 40,000–60,000 | 14.00 ± 2.00 | |
| **Number of family members** | | 0.733 |
| < 5 | 15.00 ± 7.00 | |
| >= 5 | 14.00 ± 8.00 | |

Note:
"c" represents the independent sample median test.

The United States Food and Drug Administration (FDA) has recommended take back systems such as returning medicines to the pharmacies for safe disposal, as proper methods to be used by households [28]. Compared to the current study the situation was different in countries like Malaysia, where nearly all participants (93%) knew about the drug take-back systems [29]. The evidence also shows that participants from India knew better about the system for returning expired medicines to the pharmacies [25].

Likewise, nearly 65% of the respondents "strongly agreed" that doctors and health professionals should provide advice on the safe disposal of unused and expired household medicines. The best location to educate the community about the appropriate disposal of unused medicine was identified as at the pharmacy while dispensing. The largest proportion of participants,

120(31.3%), also preferred religious places to educate the public about proper medicine disposal methods, followed by community meeting places 100(26%). However, different preferences were shown in Malaysia where 60.7% of the respondents mentioned that the best way to educate the public about the disposal of unused medication was through school, university, and public campaigns [30].

## Practice

A meta-analysis showed that the prevalence of storing medicines in houses was 77% [31]. In our study also only 88 (41.9%) participants said that they rarely store unused medications while 91.9% believed that storing excess medicines at home may promote self-medication. In Nepal like in many other developing countries, many medicines including prescription-only medicines can be purchased over the counter. While responsible self-medication can reduce the pressure on health systems inappropriate self-medication can lead to various adverse effects including the development of antimicrobial resistance [32,33]. The unsafe medicine disposal practices can also promote intoxication among the wildlife and can promote antimicrobial resistance [34]. A study from Nepal has mentioned that storage of unused and expired medicines at home can increase the risk of accidental or intentional ingestion of these substances and may create emergencies [35]. Another study from India has shown that about 87% of the community participants stored unused medicines at home [36]. This was more than in our study where only 41.9% of respondents stored unused medicines and in another study from Ethiopia, where 52.4% of participants stored medicines in their homes [37].

Results for practice showed that 125 (59.5%) participants always checked the expiry date of medicines. Good readability of the expiry dates in the medicines were responded to as always by 126 (60.0%) respondents. This finding was like a study from India, where two-thirds of the respondents always checked the expiry dates of the medicines [30]. The participants believed that the medicine becomes toxic if used after its expiry dates [30]. Majority of participants believed that the expired medicine must be returned to the manufacturers in another study done among dental residents [25].

The labelling of medicines is very important and should mention the expiry date in an easily visible manner. This becomes more important among the elderly and those with poor literacy [34]. Sharing the knowledge about the proper disposal of medicines is an urgent and important issue and the participants suggested that the dispensing pharmacist should explain the expiry dates of all the medicines with a special emphasis on the near-to-expiry ones.

For the statement on what do you do with any quantity of purchased medicine remaining unused at your home/hostel, many 75 (35.7%), participants response was throwing it away in the household garbage. This finding is like a study done in India, where more than 75% of respondents discarded expired medicines in the garbage which eventually reached the landfill sites [36]. Another study from India revealed that dental students also stored expired medicines and the preferred way to dispose of medicine was via normal household garbage [30]. Sasus et al found that half of the population surveyed had unused, leftover, or expired medicines at home, and over 75% of the population disposed of them through the normal waste bins, which ended in the landfills or dumpsites [38].

Medicines returned to pharmacies are not refunded in Nepal as most medicines are imported from the neighboring country, India, and are discarded under the supervision of the importers [39]. Improper practices for medicine disposal are seen among the people from the Nepali community and the researchers recommend that the government initiate medicine take-back systems.

### Limitations

No physical checking of medicine storage was carried out, and the analyses and interpretation were purely based on the responses of the household heads. The reasons for storing the medicines in the houses of the participants were not studied.

## Conclusion and recommendation

Educational level was the factor affecting knowledge whereas district of residence was the factor affecting attitude in this study. People knew about the expiry of medicines and methods to safely dispose of expired medicines. However, they practiced self-medication and stored medications at home and the safe methods of medicine disposal were not known by more than half of the participants. Most participants disposed of the unused and expired medicines in household garbage and the sink. This may cause harmful effects on the environment and can also have a harmful impact on the health of the public. This study also showed that there is a good level of knowledge and practice in disposing unused medicines among the public of Kathmandu Valley, Nepal. Educational interventions may help improve awareness of proper methods of medicine disposal. Developing and promoting a chart summarizing disposal procedures of common medicines can strengthen knowledge.

The medicine takeback system can be initially implemented in community pharmacies located in major cities and information about the system be widely disseminated. Similar studies in other regions are required.

## Acknowledgments

The authors would like to thank Mr. Binod Pariyar for collecting the data from the selected households. The authors also acknowledge all the participants of the study. They thank Prof. Ian Wilson, Adjunct Professor, IMU Centre for Education for copyediting the manuscript for language and grammar.

## Author Contributions

**Conceptualization:** Nisha Jha, Sajala Kafle, Shital Bhandary, Pathiyil Ravi Shankar.

**Data curation:** Shital Bhandary.

**Formal analysis:** Shital Bhandary.

**Funding acquisition:** Nisha Jha.

**Methodology:** Nisha Jha, Sajala Kafle, Pathiyil Ravi Shankar.

**Project administration:** Nisha Jha, Sajala Kafle.

**Resources:** Nisha Jha.

**Software:** Shital Bhandary.

**Supervision:** Sajala Kafle.

**Validation:** Shital Bhandary, Pathiyil Ravi Shankar.

**Visualization:** Shital Bhandary.

**Writing – original draft:** Nisha Jha, Sajala Kafle, Pathiyil Ravi Shankar.

**Writing – review & editing:** Nisha Jha, Sajala Kafle, Shital Bhandary, Pathiyil Ravi Shankar.

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
