## [Decision Letter · Decision Letter 0]

19 Apr 2022

PONE-D-21-38637Storage and medicine disposal by households in Kathmandu Valley, Nepal.PLOS ONE

Dear Dr. Jha,

Thank you for submitting your manuscript to PLOS ONE. After careful consideration, we feel that it has merit but does not fully meet PLOS ONE’s publication criteria as it currently stands. Therefore, we invite you to submit a revised version of the manuscript that addresses the points raised during the review process.

We look forward to receiving your revised manuscript.

Kind regards,

Francesca Baratta, PharmD, PhD

Academic Editor

PLOS ONE

“The study received funding of Nepalese rupees 125000 from the Nepal Health Research Council”

“PRS I have read the journal's policy and the author of this manuscript PRS have the following competing interests

Academic Editor at PLoS One”

Reviewers' comments:

Reviewer's Responses to Questions

**Comments to the Author**

1. Is the manuscript technically sound, and do the data support the conclusions?

Reviewer #1: Partly

Reviewer #2: Yes

Reviewer #3: Yes

2. Has the statistical analysis been performed appropriately and rigorously? 

Reviewer #1: Yes

Reviewer #2: No

Reviewer #3: No

3. Have the authors made all data underlying the findings in their manuscript fully available?

Reviewer #1: No

Reviewer #2: Yes

Reviewer #3: Yes

4. Is the manuscript presented in an intelligible fashion and written in standard English?

Reviewer #1: No

Reviewer #2: No

Reviewer #3: No

5. Review Comments to the Author

Reviewer #1: The authors assessed the knowledge, attitude and practice regarding the storage and disposal of unused and expired medicines in households of the Kathmandu valley in Nepal. I have below provide my comments and suggestions.

General

The methodology seems to be a bit unclear e.g. the sampling, of the sampling sufficient and if it is representative of the households in the Kathmandu valley. Please clarify.

Please critically revise the text across the manuscript for proper flow of ideas especially, the Introduction and Discussion sections.

Abstract

Objective: ‘to know’, please consider revising it to an action verb.

Please indicate the sample size in the abstract

The results subsection contains a lot of information on background variables. The authors could consider focusing on the main findings.

Introduction

The terms drug and medicine have been used in interchangeably throughout the manuscript. Please consider using the term medicine in the manuscript consistently.

In the introduction section, many of the paragraphs lack coherence with disjointed sentences. Please consider revising the introduction section with clear flow of in each of the paragraphs.

Page 7, line 108-110, please revise or possibly take it out as it seems to be a repetition.

Methodology

Please consider subdividing the methodology section into subsections:

Study area and period

Study design

Sampling

Here please clearly present the sample size determination aspect and the sampling technique aspects separately.

Sample size determination:

Please clarify how the sample size 210 was demined/calculated? Considering a multistage sampling was employed have the authors taken design effect in the sample size calculation? Please explain.

Sampling technique:

In regard to the sampling technique, in selecting the 10 households from each cluster please clarify what criterion was used to select a specific household (as the term quota sampling indicates nonprobability sampling).

Data collection and management

Please provide the questionnaire used in the study

Data entry analysis and management

Ethical considerations

Results

Please revise the sentence ‘Table 1 shows that around half of the respondents were from Kathmandu district whereas nearly 205 of respondents were from Bhaktapur and 30% respondents were from Lalitpur.’

How representative is the sample in the present study to the population of Kathmandu valley?

Although about half of the participants were described to have a university degree or higher the occupational categories represent mainly manual labor? Could the authors please explain this mismatch?

The authors could consider providing a more detailed age categorization.

The questions employed to assess the knowledge of participants (as presented in Table 2) are not clear (e.g. Q1 ‘Expiry date’: what aspect of this concept was measured?) what aspects were measured. Please consider making them clear.

Page 13, ‘Similarly, nearly 9 out of 10 respondents mentioned that the storage of excess medicines at home may promote self-medication.’ Can the authors clarify the perspective used in asking this question? Self-medication could involve prescription-only or OTC medicines. What view point was used in asking this question?

Page 13, lines 193 to 199 seems to belong in the Methodology section. Please consider moving it.

Page 13-14, lines 200-205, please explain how the different mean values were calculated, in the Methodology section.

In the ANOVA test for the mean knowledge score across educational levels please consider collapsing some of the groups to get sufficient numbers per cell. E.g. 1: primary or no formal education, 2: secondary education and 3: university education

Please explain the superscript in Table 5 (District, 0.026c).

What is the-value cut-off used to determine statistical significance. Please indicate (as in the Monthly income 0.087 in Table 5 seems to be considered significant)?

Page 21, line 238-241: was the interpretation based on a post hoc test? Please clarify

Discussion

Please provide a short summary of the main findings of the study in the first paragraph.

The discussion seems to focus on the practice aspect and the organization seems to lack flow. I think the authors could consider revising it in a way that discussion of the main findings on knowledge and attitude presented discussed in comparison with findings from other studies comes before the part about the practice aspect. The adjustment of the flow and presentation of the idea accordingly will be helpful to readers to follow the story message of the study findings in the context if previous studies.

Page 23, lines 84 to 89, the text provided seems to be out of place please consider moving this to the introduction or revise it in a way that it fits in to the idea of the discussion in a better way.

Reviewer #2: The research did not consider the design effect for sample size determination and the sampling technique was needing a probability method to include households in the study. The discussion should focus on the findings and discuss in comparison to other literatures.

Reviewer #3: Line 112 - "Cross-Sectional" is repeated twice. Kindly revise

Results: Information detailed in Table 1 are repeated extensively in-text. No need for the repetition

Table 1: Table to be formatted according to the journal's specification

Lines 188-192: Repetition of information detailed in Table 2. Kindly revise

Table 4: Not sure if the mean is the appropriate measure of central tendency here. The median has been postulated especially when using a Likert scale. The authors may want to explain their choice of mean. Also the authors to check that Table is formatted according to journal's specification

Lines 217-225: There is no need to repeat the information in the table in-text. One may highlight one or two important ones and then refer to the table.

Line 226: Same comment about the use of mean/median

Line 232: The more reason why should not be considered here at all

Table 5: please indicate the meaning of "c" in 0.026c as a footnote

Line 253: "Maximum respondents" - Authors may consider using "majority of the respondents"

Lines 280-289: The segment is not too relevant to the findings of the study. The authors may want to delete.

Line 293: Full stop(.) after "self-medication"

Lines 301-304: Using medical and nursing students for comparison here may not be too appropriate because of their knowledge base about medicines versus the general public

Lines 335-337: This statement can be re-phrased for better clarity

Lines 367-368: This part of the conclusion does not reflect results of the study - Please see Line 252 -"The safe methods of medicine disposal were not known by many, 137 (65.2%) participants". The authors to kindly revise this section of the manuscript.

Line 440 (reference no 20: "Organization WH"- To be corrected -World Health Organization

6. PLOS authors have the option to publish the peer review history of their article (what does this mean?). If published, this will include your full peer review and any attached files.

Reviewer #1: No

Reviewer #2: No

Reviewer #3: **Yes: **Prof Joseph O. Fadare

---

## [Author Response · Author response to Decision Letter 0]

24 May 2022

Revision letter

Lalitpur, Nepal

Date: 23rd May 2022

To

The Editor-in-Chief

PLOS ONE

Sub: Submission of the revised version of the manuscript for consideration of publication 

Dear Editor-in-Chief,

We are resubmitting the manuscript “Assessment of knowledge, attitude, and practice of disposing and storing of unused and expired medicines among the communities of Nepal.” after revising it and making necessary corrections for consideration of publication in your esteemed journal. We are grateful to the reviewers for their constructive comments. 

The amended Role of Funder statement in the cover letter is: "The funders had no role in study design, data collection, and analysis, decision to publish, or preparation of the manuscript."

The response to specific comments is as follows: 

General comments

The write of the manuscript has grammatical, spelling and punctuation errors to be corrected.

Response: Thank you. We have corrected it as suggested.

Specific comments

1. Title need to be modified as ‘Assessment of knowledge, attitude and practice of disposing and storing of unused and expired medicines among the communities of Nepal’.

Response: Thank you. We have corrected it as suggested.

2. Method

• Clustering is applied for multistage sampling techniques of large population. Why multi-staging was applied for your research? How many households are in each cluster?

Response: Kathmandu valley has a large population and there are 21 municipalities. We needed to represent households of these 21 municipalities and thus we used the multi-staged cluster sampling approach. We selected 10 households from each cluster to fulfill the sample size obtained for the two-stage cluster survey. The 21 clusters were selected using simple random sampling methods from all the clusters (wards) listed in each of the municipalities. There were minimum number of 1800 households in Mahankal Gaupalika of Lalitpur district and maximum number of 82336 households in the Kathmandu Metropolitan City of Kathmandu districts as per 2011 census. The number of households in each cluster were as low as 200 and as high as 2576. 

• The quota sampling technique is not appropriate for selecting the households as there will be different size households in each cluster. It would have been good to use probability method of selecting the households in each cluster as the method of sampling and then sample size could be increased to increase representativeness of the data. 

Response: We calculated the sample size for the study and based on it we needed around 210 households. Thus, we decided to select 10 households randomly from each of the randomly selected clusters. We agree that a proportional allocation would have been better, but it would not give a representative sample size for municipalities with small number of households for this study. We could have increased the sample size using a different margin of error but the budget from the funding agency was not sufficient to include a greater number of respondents.

• The sample size determination would have been considering the design effect of multistage design to increase sample size. Why you did not consider it? Why did you determine to take 10 households from each cluster? It would have been good to include more households in the study.

Response: We have used the design effect of 2 in the sample size calculation and we also added a 5% non-response error thereafter. Details are provided below as minimum sample size without these adjustments was 97 only. 

• Make it clear whether the pilot study was conducted on the households under the cluster of interest. 

Response: Yes, we conducted the pilot test on the households under the cluster of interest in Lalitpur district. 

• It would have been great to collect practice data, at least better to be supported, by observation method of data collection in each household.

Response: We used the questionnaire to obtain the “self-declared” data on practice; our study design was not based on observation. This is a very good suggestion, and we will use it if we do a similar study with a larger sample size in the future.

3. Results

• The description of Table 1 is not in agreement with the one presented in the table (205 versus 19% of respondents in Bhaktapur cluster)

Response: The total is 205 and 40 households were from Bhaktapur district as it has 4 municipalities only. Kathmandu has 11 and Lalitpur has 6 municipalities so there are 110 samples from Kathmandu and 60 samples from the Lalitpur districts of Kathmandu valley. This has been mentioned in the methods on page 11 line 205.

• After finishing the description of the findings, you have to put the referring table in the bracket as (Table …)

Response: Thank you. We have corrected it as suggested. Page 14, line 217.

• The overall description of the findings of practice of disposing and storing medicines has redundancy. So, put it in summarized form

Response: This has been edited as per the suggestion. 

4. Discussion

• The introduction part of the discussion is not related to your findings rather focus on health insurance and about WHO guidelines. Therefore, please edit and discuss based on the findings you got in a clear and brief manner by comparing your findings with other similar literatures with justification.

Response: This has been edited and the details of the health insurance and WHO guidelines has been deleted. Findings have been discussed with other similar literature with justification.

• Based on the statistical analysis of knowledge and attitude, which group of people will be the target group for intervention strategies and discuss them very well.

Response: Knowledge scores were less for the participants from Kathmandu, age more than 30 years, males, illiterates, participants having none or primary level study, housewives, people having less than 10000 NPR (80.65 USD) and having less than 5 family members. Similarly, the attitude scores showed less scores for participants from Kathmandu and Bhaktapur, having age of more than 30 years, males, literates, with primary educational level, belonging to daily wage, service, and housewife groups, having monthly income NPR 20000-40000 and 40000-60000 (162-323 and 323-484 USD) and participants with more than 5 family members. This has been added in discussion section. Page number 24, Line number 320-328.

Response: We have formatted the manuscript as per the guidelines of the journal.

Response: The grant number is 2554, 24th March 2021, Funding Agency is the Nepal Health Research Council. 

“The study received funding of Nepalese rupees 125000 from the Nepal Health Research Council”

Response: The amended Role of Funder statement in the cover letter is: "The funders had no role in study design, data collection, and analysis, decision to publish, or preparation of the manuscript."

Response: Thank you. 

“PRS I have read the journal's policy and the author of this manuscript PRS has the following competing interests

Academic Editor at PLoS One”

Response: Updated Competing Interests statement in our cover letter mentions "This does not alter our adherence to PLOS ONE policies on sharing data and materials.”

Reviewers' comments:

Reviewer's Responses to Questions

Comments to the Author

1. Is the manuscript technically sound, and do the data support the conclusions?

Reviewer #1: Partly

Reviewer #2: Yes

Reviewer #3: Yes

Response: Thank you

2. Has the statistical analysis been performed appropriately and rigorously?

Reviewer #1: Yes

Reviewer #2: No

Reviewer #3: No

Response: Thank you

3. Have the authors made all data underlying the findings in their manuscript fully available?

Reviewer #1: No

Reviewer #2: Yes

Reviewer #3: Yes

Response: Thank you

4. Is the manuscript presented in an intelligible fashion and written in standard English?

Reviewer #1: No

Reviewer #2: No

Reviewer #3: No

Response: We have copyedited the manuscript to improve the standard of written English. 

5. Review Comments to the Author

Reviewer #1: The authors assessed the knowledge, attitude and practice regarding the storage and disposal of unused and expired medicines in households of the Kathmandu valley in Nepal. I have below provide my comments and suggestions.

General

The methodology seems to be a bit unclear e.g. the sampling, of the sampling sufficient and if it is representative of the households in the Kathmandu valley. Please clarify.

Response: - We have used a sampling method that is representative of the households of the Kathmandu valley. The sample size is small due to the selection of one ward (or cluster) of the 21 municipalities to conduct the study within the budget provided.

Please critically revise the text across the manuscript for proper flow of ideas especially, the Introduction and Discussion sections.

Response: This has been carried out. 

Abstract

Objective: ‘to know’, please consider revising it to an action verb.

Response: – We agree and are sorry for not catching this earlier. We have made the change requested. Page 3, line number 39.

Please indicate the sample size in the abstract

Response: Thank you. We have added the sample size in the abstract. Page 3, line number 43.

The results subsection contains a lot of information on background variables. The authors could consider focusing on the main findings.

Response: Thank you. We have focused on the main findings as suggested. Page 3, line number 49-56.

Introduction

The terms drug and medicine have been used in interchangeably throughout the manuscript. Please consider using the term medicine in the manuscript consistently.

Response: We have changed the term “drug” to “medicine” throughout the manuscript as suggested. 

In the introduction section, many of the paragraphs lack coherence with disjointed sentences. Please consider revising the introduction section with clear flow of in each of the paragraphs.

Response: The Introduction section has been edited and the required corrections carried out. 

Page 7, line 108-110, please revise or possibly take it out as it seems to be a repetition.

Response: We have taken out these sentences. 

Methodology

Please consider subdividing the methodology section into subsections:

Study area and period

Response: We have added the study area and period as suggested. Page 6, line number 105-106.

Study design

Response: Has been added separately as suggested. Page 7, line number 107.

Sampling

Here please clearly present the sample size determination aspect and the sampling technique aspects separately.

Response: This has been presented separately as suggested. Page 7, line number 113-129.

Sample size determination:

Please clarify how the sample size 210 was demined/calculated? Considering a multistage sampling was employed have the authors taken design effect in the sample size calculation? Please explain.

Response: There was one published study from Pokhara (Western Nepal) showing 50% of the respondents doing the proper practice of drug disposal and we have provided the reference in the revised manuscript. We took this as the reference to calculate the sample size for this study. As such a study was not done earlier in Kathmandu valley, we decided to use a margin of error of 10% giving a proper practice range of 40-60% with a 95% confidence interval for the sample size. We used www.openepi.com to get the sample size of 97 using these parameters. This sample size was multiplied by a design effect of 2 to adjust the variance introduced due to clusters (random wards) giving a sample size of 193, which was further adjusted with a 5% non-response rate giving the final minimum sample size of 203. As we planned to select an equal number of households from the one randomly selected ward of each of the 21 municipalities of the Kathmandu Valley, we decided to randomly select 10 households each from each municipality giving the final sample size of 210 for this study. The ward was selected using the “randbetween” function in using the number of wards in each municipality as the frame whereas households were selected using systematic random sampling in the sampling frame obtained from the selected ward (cluster) office. This was a two-stage cluster sampling with random sampling in both stages.

Sampling technique:

In regard to the sampling technique, in selecting the 10 households from each cluster please clarify what criterion was used to select a specific household (as the term quota sampling indicates nonprobability sampling).

Response: The research assistant hired and trained for this study identified the ward office and handed over the ethical approval letter from the Nepal Health Research Council there. Once he got the administrative approval, he randomly selected the 10 households using the sampling frame provided by the ward office. He then proceeded with the data collection from these 10 households. We decided to select 10 households from each of 21 municipalities so fulfill the minimum sample size for the two-stage cluster survey design used in this study. 

Data entry analysis and management

Response: The collected questionnaires were first assigned with the unique id, and it was checked manually by the study team. Once the study team was satisfied with the filled questionnaire, then it was entered directly into the pre-finalized SPSS file by one of the study members. Once all the data were entered, it was checked with frequencies and any inconsistencies found were corrected as per the unique id and the corresponding questionnaire by the study team. The final data was saved securely on the principal investigator’s personal laptop. A working copy with the de-identified file was used for the data analysis. 

Please provide the questionnaire used in the study

Response: This has been added as a supplementary file. 

Data entry analysis and management

Response: Has been mentioned as suggested. 

Ethical considerations

Response: has been added as suggested. 

Results

Please revise the sentence ‘Table 1 shows that around half of the respondents were from Kathmandu district whereas nearly 205 of respondents were from Bhaktapur and 30% respondents were from Lalitpur.’

Response: This has been done. Table 1 shows that around half, 110 (52.4%) of the respondents were from Kathmandu district whereas 40 (19.5%) of respondents were from Bhaktapur and 60 (28.6%) respondents were from Lalitpur.

Page 11, line number 203-205.

How representative is the sample in the present study to the population of Kathmandu valley?

Response: We have used a two-stage cluster sampling and the study population represents all the households of the Kathmandu valley.

Although about half of the participants were described to have a university degree or higher the occupational categories represent mainly manual labor? Could the authors please explain this mismatch?

Response: This may reflect the current economic situation in Nepal. Many persons with a degree and other educational qualifications may not be obtaining a job commensurate with their qualifications and may be involved in other occupational categories including manual labor. This was not an objective of our study, so we did not investigate it further. 

The authors could consider providing a more detailed age categorization.

Response: The age, a continuous variable, was categorized into two categories using median as the cut-off value. We used the median to divide the whole data into two parts as the median is not prone to the presence of outliers and extreme values.

The questions employed to assess the knowledge of participants (as presented in Table 2) are not clear (e.g. Q1 ‘Expiry date’: what aspect of this concept was measured?) what aspects were measured. Please consider making them clear.

Response: We have provided the complete statements in table 2.

Page 13, ‘Similarly, nearly 9 out of 10 respondents mentioned that the storage of excess medicines at home may promote self-medication.’ Can the authors clarify the perspective used in asking this question? Self-medication could involve prescription-only or OTC medicines. What viewpoint was used in asking this question?

Response: In Nepal like in many other developing countries many medicines including prescription-only medicines can be purchased over the counter. While responsible self-medication can reduce the pressure on health systems inappropriate self-medication can lead to various adverse effects including the development of antimicrobial resistance. Page 23, line number 293-297.

Page 13, lines 193 to 199 seems to belong in the Methodology section. Please consider moving it.

Response: Yes, this sentence has been moved to methodology as suggested. Page 10, line number 174-176.

Page 13-14, lines 200-205, please explain how the different mean values were calculated, in the Methodology section.

Response: It has been added in the methodology section.

The first mean and 95% confidence errors were computed using a simple random sampling approach while the second mean and its 95% confidence error were computed using a complex/cluster sampling analysis plan in IBM SPSS software as we have used a two-stage cluster survey design in this study.

In the ANOVA test for the mean knowledge score across educational levels please consider collapsing some of the groups to get sufficient numbers per cell. E.g. 1: primary or no formal education, 2: secondary education and 3: university education

Response: We agree and have presented the modified analysis accordingly.

Descriptives

Knowledge scale 

 N Mean Std. Deviation Std. Error 95% Confidence Interval for Mean Minimum Maximum

 Lower Bound Upper Bound 

1 24 5.54 1.318 .269 4.99 6.10 2 7

2 85 5.51 1.151 .125 5.26 5.75 3 8

3 101 5.97 1.063 .106 5.76 6.18 3 8

Total 210 5.73 1.147 .079 5.58 5.89 2 8

ANOVA

Knowledge scale 

 Sum of Squares df Mean Square F Sig.

Between Groups 10.950 2 5.475 4.291 .015

Within Groups 264.116 207 1.276 

Total 275.067 209 

The p-value is still significant!

Please explain the superscript in Table 5 (District, 0.026c).

Response:

- The title of this table is not correct. It has been corrected as “Comparison of attitude score among different subgroups of respondents”

- The “c” represents the independent sample median test, and it is added to the bottom of table 5.

- 

What is the-value cut-off used to determine statistical significance. Please indicate (as in the Monthly income 0.087 in Table 5 seems to be considered significant)?

Response: The cut-off used to determine the statistical significance was 0.05. The p-value of 0.087 of the monthly income in Table 5 was not statistically significant. We have made changes and not bolded or highlighted it now. 

Page 21, line 238-241: was the interpretation based on a post hoc test? Please clarify

Discussion

Response- Yes, it was based on the post-hoc test given by the IBM SPSS software. 

Please provide a short summary of the main findings of the study in the first paragraph.

Response-Thank you. We have provided it now in the first paragraph of the main finding. 

The discussion seems to focus on the practice aspect and the organization seems to lack flow. I think the authors could consider revising it in a way that discussion of the main findings on knowledge and attitude presented discussed in comparison with findings from other studies comes before the part about the practice aspect. The adjustment of the flow and presentation of the idea accordingly will be helpful to readers to follow the story message of the study findings in the context if previous studies.

This has been done. 

Page 23, lines 84 to 89, the text provided seems to be out of place please consider moving this to the introduction or revise it in a way that it fits in to the idea of the discussion in a better way.

Response: We have deleted these sentences as suggested by the second reviewer. 

Reviewer #2: 

The research did not consider the design effect for sample size determination and the sampling technique was needing a probability method to include households in the study. 

The discussion should focus on the findings and discuss in comparison to other literatures.

Response: Thank you. We have used a design effect of 2 while calculating the sample size and used 5% non-response rate as the sample size was 97 without these adjustments. We selected one ward (cluster) randomly from each of 21 municipalities using “randbetwen” function of excel. The research assistant compiled the sampling frame (households) in the selected clusters and selected 10 households using systematic random sampling in the sampling frame provided by the ward office after getting approval from the administrators. This was a two-stage cluster sampling design.

We have now focused on the key finding and compared and contrasted these with the literature.

Reviewer #3: Line 112 - "Cross-Sectional" is repeated twice. Kindly revise

Response: Thank you. We have done the correction now.

Results: Information detailed in Table 1 are repeated extensively in-text. No need for the repetition

The repetition has been reduced. 

Table 1: Table to be formatted according to the journal's specification

Response: Thank you. We have revised it now.

Lines 188-192: Repetition of information detailed in Table 2. Kindly revise

Response: Thank you. We have revised it now.

Table 4: Not sure if the mean is the appropriate measure of central tendency here. The median has been postulated especially when using a Likert scale. The authors may want to explain their choice of mean. Also the authors to check that Table is formatted according to journal's specification

Response: The mean and standard deviation presented here are the “weighted average” based on frequencies and the forced Likert scale as weights. This is a proxy of the “consensus index” and is used here to show the direction of the attitude. We have used the median and IQR while comparing the attitude scale with background variables in Table 5.

Thank you. We have formatted the table according to the journal’s specification. 

Lines 217-225: There is no need to repeat the information in the table in-text. One may highlight one or two important ones and then refer to the table.

Response: Thank you. We have made the suggested changes.

Line 226: Same comment about the use of mean/median

Response: When a scale is formed using a Likert item, it will have different levels of measurement i.e. interval from ordinal. Thus, we have summarized the attitude scale using mean and standard deviation. We have not used regular arithmetic mean and standard deviation to summarize the individual item, however, as they cannot be done as per their measurement scale.

Line 232: The more reason why should not be considered here at all

Response: We have treated the attitude scale as the interval scale variable and hence it was assessed with a histogram to check whether the parametric or non-parametric tests were appropriate for the summated attitude scale variable.

Table 5: please indicate the meaning of "c" in 0.026c as a footnote 

Response: The “c” means independent sample median test. 

Line 253: "Maximum respondents" - Authors may consider using "majority of the respondents"

Response: Thank you. We have changed as per the suggestion now.

Lines 280-289: The segment is not too relevant to the findings of the study. The authors may want to delete.

Response: Thank you. We have deleted it from the text.

Line 293: Full stop(.) after "self-medication"

Response: Thank you. We have added the full stop now.

Lines 301-304: Using medical and nursing students for comparison here may not be too appropriate because of their knowledge base about medicines versus the general public

Response: Thank you. We have modified the text as per the suggestion. More studies on medicine disposal among the general public has been added as suggested. Page 23, line number 301 to 306.

1. Nepal S, Giri A, Bhandari R. Poor and Unsatisfactory Disposal of Expired and Unused Pharmaceuticals: A Global Issue. Curr Drug Saf. 2020;15(3):167-172. 

2. Manocha S, Suranagi UD, Sah RK. Current Disposal Practices of Unused and Expired Medicines Among General Public in Delhi and National Capital Region, India. Curr Drug Saf. 2020;15(1):13-19. 

3. Kahsay H, Ahmedin M, Kebede B, Gebrezihar K, Araya H, Tesfay D. Assessment of Knowledge, Attitude, and Disposal Practice of Unused and Expired Pharmaceuticals in Community of Adigrat City, Northern Ethiopia. J Environ Public Health. 2020;2020:6725423. 

Lines 335-337: This statement can be re-phrased for better clarity

Response: Thank you. We have re-phrased the statement now.

Lines 367-368: This part of the conclusion does not reflect results of the study - Please see Line 252 -"The safe methods of medicine disposal were not known by many, 137 (65.2%) participants". The authors to kindly revise this section of the manuscript.

Response: Thank you. We have revised this section of the manuscript.

Line 440 (reference no 20: "Organization WH"- To be corrected -World Health Organization

Response: Thank you. We have corrected reference number 20 as suggested. 

6. PLOS authors have the option to publish the peer review history of their article (what does this mean?). If published, this will include your full peer review and any attached files.

Do you want your identity to be public for this peer review? For information about this choice, including consent withdrawal, please see our Privacy Policy.

Reviewer #1: No

Reviewer #2: No

Reviewer #3: Yes: Prof Joseph O. Fadare

New references have been added and several references have been renumbered. The changes suggested by the reviewers have been accepted and carried out in the manuscript using red font. All cited references are accurate. The revisions have been approved by all the authors. 

Hoping for a favorable consideration

Thanking you

Yours Sincerely,

Nisha Jha and coauthors 

Professor, Department of Pharmacology 

KIST Medical College

Imadol, Lalitpur, Nepal 

E-mail: nishajha32@gmail.com

Phone: 00977-01-5201680 

Fax: 00977-01-5201496

---

## [Decision Letter · Decision Letter 1]

14 Jun 2022

PONE-D-21-38637R1Assessment of knowledge, attitude, and practice of disposing and storing of unused and expired medicines among the communities of Kathmandu, Nepal.PLOS ONE

Dear Dr. Jha,

Thank you for submitting your manuscript to PLOS ONE. After careful consideration, we feel that it has merit but does not fully meet PLOS ONE’s publication criteria as it currently stands. Therefore, we invite you to submit a revised version of the manuscript that addresses the points raised during the review process.

We look forward to receiving your revised manuscript.

Kind regards,

Francesca Baratta, PharmD, PhD

Academic Editor

PLOS ONE

Journal Requirements:

Reviewers' comments:

Reviewer's Responses to Questions

**Comments to the Author**

1. If the authors have adequately addressed your comments raised in a previous round of review and you feel that this manuscript is now acceptable for publication, you may indicate that here to bypass the “Comments to the Author” section, enter your conflict of interest statement in the “Confidential to Editor” section, and submit your "Accept" recommendation.

Reviewer #1: (No Response)

Reviewer #3: (No Response)

2. Is the manuscript technically sound, and do the data support the conclusions?

Reviewer #1: Partly

Reviewer #3: Yes

3. Has the statistical analysis been performed appropriately and rigorously? 

Reviewer #1: No

Reviewer #3: Yes

4. Have the authors made all data underlying the findings in their manuscript fully available?

Reviewer #1: Yes

Reviewer #3: Yes

5. Is the manuscript presented in an intelligible fashion and written in standard English?

Reviewer #1: No

Reviewer #3: No

6. Review Comments to the Author

Reviewer #1: Review report

I would like to thank the authors for making revisions to the manuscript based on reviewers’ comments. However, I believe important issues remain to be addressed. I have listed some comments and suggestions which I believe will help improve the manuscript further.

Abstract

- In page 55 of the manuscript file, on page 3 line 39 to 41, ‘The study explored the status of medicine disposal and storage practices and evaluated knowledge, attitude, and practice of medicine storage and disposal techniques among households of the three districts of the Kathmandu valley, Nepal.’ Please revise the sentence, as it stands it seems a bit confusing and contains repetitions.

- In page 55 of the manuscript file, on page 3, the sentences in Results subsection ‘The safe methods of medicine disposal were not known by 137 (65.2%) participants. Throwing in a dustbin was the preferred method of expired medicine disposal.’ Seem to be contradictory to the conclusion subsection ‘There is good level of knowledge and practice 58 of disposing the unused medicines among the public of Kathmandu Valley’ please explain why or correct the conclusion

Introduction

- At the end of the introduction section the objective stated ‘This study evaluates medicine disposal and storage practices and knowledge, attitude, and practice of medicine storage and disposal techniques among households residing in the three districts of the Kathmandu valley.’ Please make the objective presented at the end of the introduction section the same as the one in the abstract (of course after revising the sentence to be concise and clear)

Methodology

- ‘study area and period’: in this subsection the authors could consider providing a short introduction to Kathmandu to readers (as the readership of the journal is international) who may not be familiar with the area. For example, the authors could describe the general socio-demographic and health information of the area.

- Was there a cut-off score to identify a respondent considered knowledgeable on medicine storage and disposal?

- Have the authors considered conducting a regression analysis (for the knowledge and attitude scores) to assess the factors associated to them in a manner that will take account of potential confounders? This is crucial in determining which factors are really responsible for/predict knowledge or attitude after controlling for other potential confounders.

Results

- Please use subheadings to identify sections such as ‘Demographic characteristics’, ‘Knowledge’, ‘Attitude’ and ‘Practice’. This would help o low o ideas and for ease of following by readers.

Discussion

- I believe the discussion section still requires more work in terms of structure and flow of ideas. I find it difficult to follow the thread of thought in this section. In my opinion, starting by providing a summary of the main findings in the first paragraph and in the following paragraphs expounding on these findings by comparing with other studies and discussing implications might improve this section. In terms of order the themes and findings within knowledge, attitude and practice components could be presented in the sequential paragraphs.

Conclusion

- I notice a similar issue in the Conclusion where practice issues are presented before addressing knowledge.

- Please address this and be consistent in terms of ideas (i.e. I believe the findings should inform the discussion and conclusions).

Reviewer #3: Most of the suggested revisions were not carried out by the authors.

Examples:

Lines 301 -304 : Using medical and nursing students for comparison here may not be too appropriate because of their knowledge base about medicines versus the general public

Lines 335-337: This statement can be re-phrased for better clarity

Lines 217-223: There is no need to repeat the information in the table in-text. One may highlight one or two important ones and then refer to the table.

The tables to conform to the journal's specifications

Overall, the standard of English language needs to be improved upon

7. PLOS authors have the option to publish the peer review history of their article (what does this mean?). If published, this will include your full peer review and any attached files.

Reviewer #1: No

Reviewer #3: **Yes: **Prof. Joseph O. Fadare

---

## [Author Response · Author response to Decision Letter 1]

21 Jul 2022

Revision letter

Lalitpur, Nepal

Date: 22nd July 2022

To

The Editor-in-Chief

PLOS ONE

Sub: Submission of the revised version of the manuscript for consideration of publication 

Dear Editor-in-Chief,

We are resubmitting the manuscript “Assessment of knowledge, attitude, and practice of disposing and storing unused and expired medicines among the communities of Nepal.” after revising it and making necessary corrections for consideration for publication in your esteemed journal. We are grateful to the reviewers for their constructive comments. 

The response to specific comments is as follows: 

Abstract

Comment 1- In page 55 of the manuscript file, on page 3 line 39 to 41, ‘The study explored the status of medicine disposal and storage practices and evaluated knowledge, attitude, and practice of medicine storage and disposal techniques among households of the three districts of the Kathmandu valley, Nepal.’ Please revise the sentence, as it stands it seems a bit confusing and contains repetitions.

Response: This has been corrected as: 

The study evaluated knowledge, and attitude regarding unused and expired medicines and explored medicine storage and disposal practices among selected households in the Kathmandu valley, Nepal.

Comment 2- In page 55 of the manuscript file, on page 3, the sentences in Results subsection ‘The safe methods of medicine disposal were not known by 137 (65.2%) participants. Throwing in a dustbin was the preferred method of expired medicine disposal.’ Seem to be contradictory to the conclusion subsection ‘There is good level of knowledge and practice 58 of disposing the unused medicines among the public of Kathmandu Valley’ please explain why or correct the conclusion

Response: The statement has been corrected in the conclusion part as stated below.

The level of knowledge and practice of disposing of unused and expired medicines requires improvement.

Introduction

Comment 3 - At the end of the introduction section the objective stated ‘This study evaluates medicine disposal and storage practices and knowledge, attitude, and practice of medicine storage and disposal techniques among households residing in the three districts of the Kathmandu valley.’ Please make the objective presented at the end of the introduction section the same as the one in the abstract (of course after revising the sentence to be concise and clear)

Response: This has been changed as suggested.

The study evaluated knowledge, and attitude regarding unused and expired medicines and explored medicine storage and disposal practices among selected households in the Kathmandu valley, Nepal.

Methodology

Comment 4- ‘study area and period’: in this subsection the authors could consider providing a short introduction to Kathmandu to readers (as the readership of the journal is international) who may not be familiar with the area. For example, the authors could describe the general socio-demographic and health information of the area.

Response: A description of Kathmandu has been added as suggested at the beginning of the Methodology section.

Comment 5 - Was there a cut-off score to identify a respondent considered knowledgeable on medicine storage and disposal?

Response: We did not use any cut-off score to classify the knowledge score.

Comment 6- Have the authors considered conducting a regression analysis (for the knowledge and attitude scores) to assess the factors associated to them in a manner that will take account of potential confounders? This is crucial in determining which factors are really responsible for/predict knowledge or attitude after controlling for other potential confounders.

Response: Thank you. Since the knowledge score was only significantly different according to one of the socio-economic variables on the t-test/1-way ANOVA and other variables had a p-value > 0.25, it did not make sense to proceed with the multivariate analysis to find an independent effect controlling for confounders using regression analysis. Attitude score was found to be skewed and so an independent samples median test was used to compare it across socio-demographic variables, linear regression was not applicable for this data and thus it was not considered for the multivariate analysis.

Results

Comment 7- Please use subheadings to identify sections such as ‘Demographic characteristics’, ‘Knowledge’, ‘Attitude’ and ‘Practice’. This would help o low o ideas and for ease of following by readers.

Response: This has been added as suggested. 

Discussion

Comment 8- I believe the discussion section still requires more work in terms of structure and flow of ideas. I find it difficult to follow the thread of thought in this section. In my opinion, starting by providing a summary of the main findings in the first paragraph and in the following paragraphs expounding on these findings by comparing with other studies and discussing implications might improve this section. In terms of order the themes and findings within knowledge, attitude and practice components could be presented in the sequential paragraphs.

Response: This has been corrected as suggested. 

Conclusion

Comment 9- I notice a similar issue in the Conclusion where practice issues are presented before addressing knowledge.

Response: This has been addressed as suggested. 

Comment 10- Please address this and be consistent in terms of ideas (i.e. I believe the findings should inform the discussion and conclusions).

Response: Thank you. This has been corrected as suggested. 

Reviewer #3: Most of the suggested revisions were not carried out by the authors.

Examples:

Comment 11- Lines 301 -304 : Using medical and nursing students for comparison here may not be too appropriate because of their knowledge base about medicines versus the general public

Response: Thank you for the comment. The newly added references are for comparing the results with the general public and not with the nursing and medical students. 

Comment 12- Lines 335-337: This statement can be re-phrased for better clarity

Response: This has been rephrased as: Another study mentioned that only 13.3% of respondents were aware of the proper medicine disposal methods and only 1% of participants knew about returning the medicines to the pharmacies for the medicine take-back systems.25

Comment 13- Lines 217-223: There is no need to repeat the information in the table in-text. One may highlight one or two important ones and then refer to the table.

Response: The corrected part is written as: The knowledge scores were significantly different only among respondents with different educational levels (p-value < 0.05). So, a pairwise comparison was done for this variable using Tukey’s HSD post-hoc test, and a statistically significant difference was observed between respondents with Secondary and University levels only (Table 3).

Comment 14- The tables to conform to the journal's specifications

Response: The tables are as per the guidelines of the journal.

Comment 15- Overall, the standard of English language needs to be improved upon

Response: The manuscript has been copyedited for language issues. 

7. PLOS authors have the option to publish the peer review history of their article (what does this mean?). If published, this will include your full peer review and any attached files.

Do you want your identity to be public for this peer review? For information about this choice, including consent withdrawal, please see our Privacy Policy.

Reviewer #1: No

Reviewer #3: Yes: Prof. Joseph O. Fadare

Hoping for a favorable consideration

Thanking you

Yours Sincerely,

Nisha Jha and coauthors 

Professor, Department of Pharmacology 

KIST Medical College

Imadol, Lalitpur, Nepal 

E-mail: nishajha32@gmail.com

Phone: 00977-01-5201680 

Fax: 00977-01-5201496

---

## [Editor Report · Decision Letter 2]

25 Jul 2022

Assessment of knowledge, attitude, and practice of disposing and storing unused and expired medicines among the communities of Kathmandu, Nepal.

PONE-D-21-38637R2

Dear Dr. Jha,

We’re pleased to inform you that your manuscript has been judged scientifically suitable for publication and will be formally accepted for publication once it meets all outstanding technical requirements.

Kind regards,

Francesca Baratta, PharmD, PhD

Academic Editor

PLOS ONE

---

## [Editor Report · Acceptance letter]

27 Jul 2022

PONE-D-21-38637R2 

Assessment of knowledge, attitude, and practice of disposing and storing unused and expired medicines among the communities of Kathmandu, Nepal. 

Dear Dr. Jha:

I'm pleased to inform you that your manuscript has been deemed suitable for publication in PLOS ONE. Congratulations! Your manuscript is now with our production department. 

Kind regards, 

on behalf of

Dr. Francesca Baratta 

Academic Editor

PLOS ONE